# Scots Pine Seedlings Growth Dynamics Data Reveals Properties for the Future Proof of Seed Coat Color Grading Conjecture

**Arthur Novikov [1],\*** , **Vladan Ivetić [2]** , **Tatyana Novikova [1]** and **Evgeniy Petrishchev [1]**

[1]   Mechanical Department, Voronezh State University of Forestry and Technologies named after G.F. Morozov, 8, Timiryazeva, Voronezh 394087, Russia

[2]   Faculty of Forestry, University of Belgrade, Kneza Višeslava 1, Belgrade 11030, Serbia

\*   Correspondence: arthur.novikov@vglta.vrn.ru; Tel.: +7-903-650-84-09

**Abstract:** Seed coat color grading conjecture is also known as Pravdin's conjecture. To verify the conjecture, we established a long-term field experiment. This data set included unique empirical data of Scots pine (*Pinus sylvestris* L.) container-grown seedlings produced from different seed color grades, outplanted on a post fire site in the Voronezh region, Russia. Variables were provided for 10 rows of 90 samples in each row. These data contribute to our understanding of seed germination and seedlings growth dynamics from size and color gradings of seeds. This structure is the future basis of the Forest Reproductive Material Library (FRMLib) and will be used for assisted migration and forest seed transfer.

**Dataset:** Dataset access at http://dx.doi.org/10.17632/fx4wx7hj86.2

**Dataset License:** CC BY NC 3.0

**Keywords:** post fire planted dynamics; Scots pine; seed grading; seed coat color; seed size; container-grown seedlings; seed germination

---

## 1. Summary

The Pravdin's conjecture is a significant difference in the "morphology of chromosomes in karyotypes of forest crops, depending on the geographical location and seed coat color" [1]. This hypothesis is the basis of the project "Development of forest seeds production with the specified characteristics" [2]. The project goal was the improvement of forest seed production for direct, both ground and aerial seeding [3,4], as well as storage, by devising a technology and designing technical means for qualitative and quantitative grading of seed.

Seed of the main forest-forming species, as an integral part of forest reproductive material (FRM) [5] is quite a valuable product, transferred by trade operations over long distances [6]. The quality of seed to a large extent determines the pace and final success of reforestation. An increase in the competitiveness of forest seed by improvement of quality attributes was one of the key steps in the development of the forest complex of Russia, set out in the Strategy [7]. This assumed active interaction of private forest users (and/or owners) with producers of forest seed that was consistent with the Russian forest legislation.

For Scots pine seeds, like the any plant species, field germination is the most important quality attribute. Field germination depends on internal factors like genetics and initial seed viability [8], and external factors like the years of crop formation, temperature conditions during germination [9] and can be improved by pre-sowing treatments like seed stratification, a proper substrate or soil

preparation and the nursery cultural practice after the sowing [10,11]. The results of a field germination test conducted in a container nursery [8] differs from the results in the bareroot nursery, especially for seeds with low viability. It has been proven that the germination of coniferous seed is closely related to the temperature conditions [9] and that it is slower and more unstable [12] compared to the seeds of agricultural crops, requiring additional costs for heating greenhouses in nurseries. The aim of this study was to test whether seed grading on size and color improved the performance of Scots pine seedlings grown in a container.

At the operational level, the sorting of seeds based on size prior to sowing is usually performed by specialized equipment for seed processing (in the case of this study it was produced by BCC AB Corp., Sweden). The usual technology practice involves sorting seeds sequentially both geometrically and gravitationally, which are both quantitative features. However, the spectrometric properties of seeds [13–15] determined by provenance [16] and determining the seed viability [17] should not be neglected either.

In uniform growing conditions like in the forest nursery, the viability of the individual seed has a decisive role in the production of the targeted number of seedlings [18]. Spectrometric properties of seed (i.e., seed color) is one of the attributes which indicates seed viability and germination rate. As a result of this, spectrometric properties of seeds should be taken into account when designing machine vision devices [19–23] for forest pre-sowing seed processing, especially for use in aerial seeding [24]. Combined with the quantitative feature of seed size, spectrometric properties are a reliable indicator of a seedlings performance. Additionally, we can recommend the grading of *Pinus sylvestris* L. seed on two size classes [25].

The data set was used in papers [24,26] and preprints [27,28]. The results showed a significant statistical difference in seed germination and the seedlings growth between groups produced from seeds with a different size and coat color.

## 2. Data Description

The data set is available from the Mendeley Data Repository [29] and cover one file (*Pinus sylvestris one-year data.xlsx*) and three folders:

- 30 Day Container-grown Seedlings (Photo 2017);
- 50 Day Container-grown Seedlings (Photo 2017);
- Seedlings in 10 rows (Photo Spring 2019).

*Pinus sylvestris one-year data.xlsx* file includes all information of the specific data. The file includes 10 Excel sheets (named 1, 2 … 10), sheet *Meteorology 2018* and sheet *Container seed germination* Supplementary File.

Sheets 1, 2, . . . , 10 present data on seedling growth (see structure in Table 1) depending on the technological features of seed sorting (see Table 2).

Sheet *Container seed germination* includes 19 columns (see dataset structure on the light group in Table 3). For each color-size group there are six columns.

Sheet *Meteorology 2018* includes five columns (see structure Table 4) adapted in http://pogoda-service.ru/climate_table.php:

- Month;
- Average temperature;
- Temperature normal ratio;
- Rainfall;
- Rainfall normal ratio.

**Table 1.** The table format of the growth dataset of Scots pine seedlings.

| Height in First Measurements Date (cm) | Height in Second Measurements Date (cm) | Height in Third Measurements Date (cm) | Height in Fourth Measurements Date (cm) | Height in Growth Season Finally Measurements Date (cm) | Root Collar Diameter (RCD) in Growth Season Finally Measurements Date (mm) |
|---|---|---|---|---|---|
| data 1 | data 1 | data 1 | data 1 | data 1 | data 1 |
| data 2 | data 2 | data 2 | data 2 | data 2 | data 2 |
| . . . | . . . | . . . | . . . | . . . | . . . |
| Data finally | Data finally | Data finally | Data finally | Data finally | Data finally |
| N | N | N | N | N | N |
| Survival, % | Survival, % | Survival, % | Survival, % | Survival, % | Survival, % |
| Mean | Mean | Mean | Mean | Mean | Mean |
| Average deviation | Average deviation | Average deviation | Average deviation | Average deviation | Average deviation |
| Variance | Variance | Variance | Variance | Variance | Variance |
| Standard deviation | Standard deviation | Standard deviation | Standard deviation | Standard deviation | Standard deviation |
| Coefficient of variation | Coefficient of variation | Coefficient of variation | Coefficient of variation | Coefficient of variation | Coefficient of variation |
| Oscillation factor | Oscillation factor | Oscillation factor | Oscillation factor | Oscillation factor | Oscillation factor |
| Asymmetry | Asymmetry | Asymmetry | Asymmetry | Asymmetry | Asymmetry |
| Kurtosis | Kurtosis | Kurtosis | Kurtosis | Kurtosis | Kurtosis |

**Table 2.** The technological features of Scots pine seed sorting.

| Excel Sheet Number | Seed Coat Color Group: Light (Wavelength of 650–715 nm and Reflectance 70–85%), Brown (650–715 nm and 50–65%) and Dark (650–715 nm and 35–45%) | Seed Size Group: Small (2.51 to 3.25 mm) and Large ( > 3.25 mm) | Number of Seeds |
|---|---|---|---|
| 1 (bulk) | non-graded | non-graded | 200 |
| 2 | 1 (light) | non-graded | 200 |
| 3 | 2 (brown) | non-graded | 200 |
| 4 | 3 (dark) | non-graded | 200 |
| 5 | 1 (light) | small | 200 |
| 6 | 1 (light) | large | 200 |
| 7 | 2 (brown) | small | 200 |
| 8 | 2 (brown) | large | 200 |
| 9 | 3 (dark) | small | 200 |
| 10 | 3 (dark) | large | 200 |

**Table 3.** The table format of the germination (%) dataset of Scots pine seeds.

| Container Number | Light Seeds (Day 30) | Light Large Seeds (Day 30) | Light Small Seeds (Day 30) | Light Seeds (Day 50) | Light Large Seeds (Day 50) | Light Small Seeds (Day 50) |
|---|---|---|---|---|---|---|
| 1 | data 1 | data 1 | data 1 | data 1 | data 1 | data 1 |
| 2 | data 2 | data 2 | data 2 | data 2 | data 2 | data 2 |
| … | … | … | … | … | … | … |
| Number finally | Data finally | Data finally | Data finally | Data finally | Data finally | Data finally |

**Table 4.** The table format of the meteorological dataset of growth season 2018.

| Month | Average Temperature, Grad C | Temperature Normal Ratio, Grad C | Rainfall (mm) | Rainfall Normal Ratio (mm) |
|---|---|---|---|---|
| data 1 | data 1 | data 1 | data 1 | data 1 |
| data 2 | data 2 | data 2 | data 2 | data 2 |
| … | … | … | … | … |
| Data finally | Data finally | Data finally | Data finally | Data finally |

## 3. Methods

### 3.1. Study Microsite

The experimental microsite was located on the post fire non-uprooting site of the left-bank forestry training center of the Voronezh State University of Forestry and Technologies, located in the Voronezh region, Russian Federation (coordinates of the nodal point: N 51°49′40.3′ E 39°21′49.7′, altitude 100.8 m a.s.l). The microsite had a rectangular shape of 405 m$^2$ and was divided into 10 rows made by the innovative plowing technique [30]. There were no large deviations from a normal in terms of temperatures during the experiment time, with average monthly temperature rising from May to July, and decreasing after the peak in July until the end of the experiment in September. As compared to the temperature, there were large deviations from the normal in terms of rainfall. In all months during the experiment time the rainfall was lower than normal, range from 62% of the normal in June to only 31% in August.

### 3.2. Seed Production (Collection and Processing)

Cones of *Pinus sylvestris* L. were collected in autumn of 2016, from selected trees in a natural forest located in the Pavlovsky district of the Voronezh region, Russian Federation (Latitude 50.462169; Longitude 40.096446, altitude 83 m a.s.l). Seeds were extracted from cones and further processed (pre-cleaning, extraction, de-winging) using the standard procedures and equipment (BCC AB, Landskrona, Sweden) at the Voronezh containerized forest nursery (Latitude 51.567094; Longitude 39.243006, altitude 105 m a.s.l). The original seedlot was placed in a storage facility in a glass bottle and kept at +5 ± 2 °C and humidity of 60%.

From the original seedlot, three random samples of 0.5 kg were extracted in May of 2017. The samples were kept for 24 hours at +20 ± 2 °C and humidity of 75%. From each sample, three seed coat color classes were separated by the use of standard photo-separator (Smart Grade LLC, Russia) with a significantly different degree of reflection in the wavelength range of 650 to 715 nm [14]. The organoleptic test [31] defined these color fractions as light (color class 1), brown (color class 2) and dark (color class 3). Additionally, these seeds could be classified in the Munsell [32] color system using the image processing Digital Color Guide Android software (DIC Corp., Tokyo, Japan). Three representative samples (Figure 1) were taken from each seed-color class.

At the end, seeds of each color class were graded by size using the sieve sorter (BCC AB Corp., Sweden) on two dimensional fractions of "small" seeds comprising of seeds whose width ranged from 2.51 to 3.25 mm and "large" seeds comprising seeds whose width exceeded 3.25 mm.

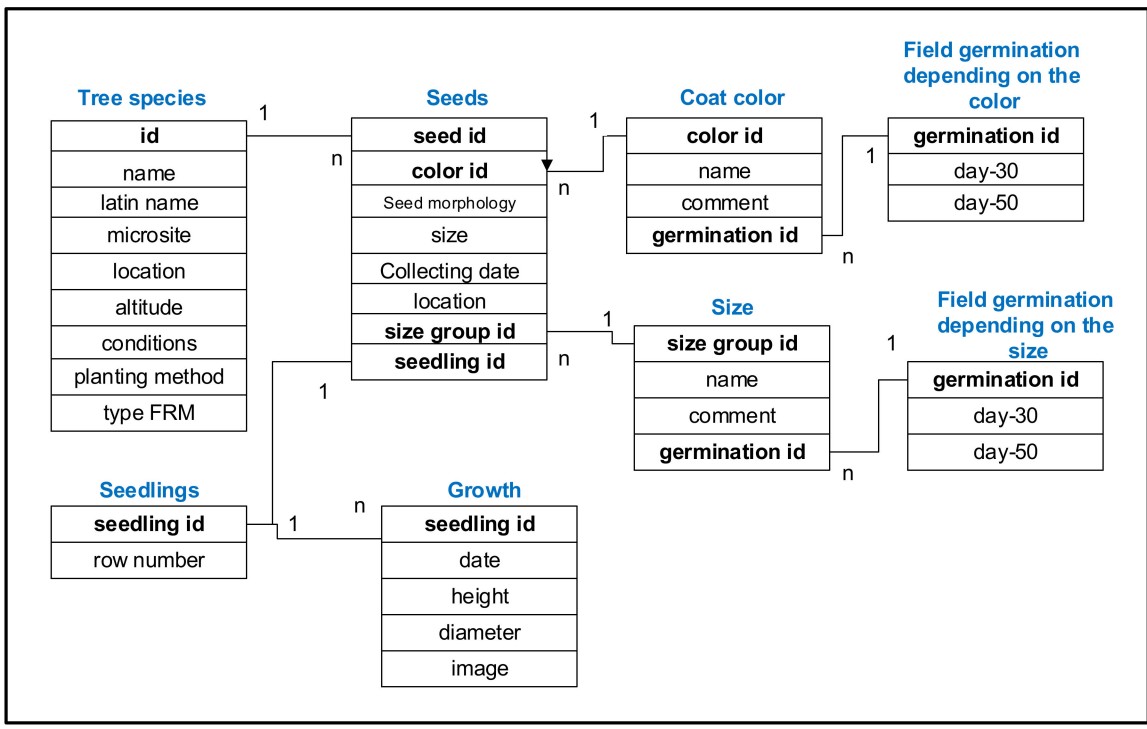

**Figure 1.** Relationship data model for image and variable collections. This structure is the future basis of the Forest Reproductive Material Library (FRMLib) and will be used for assisted migration and transfer of Forest Reproductive Material (FRM).

### 3.3. Seedlings Production

Seeds (N = 1800) of all color-size classes were sown by an automatic seeder (BCC AB Corp., Sweden) in 40-cell containers (BCC AB Corp., Sweden) filled with peat substrate. Each color group was sown in a total of five containers. Containers were installed in greenhouses with automatic maintenance of temperature and humidity. Determination of germination was performed for each container on the 30th and 50th day [33,34] from the seeding.

The resulting seedlings were removed from the container, transported in cardboard boxes and planted under the Kolesov sword in the row bottom on October 24, 2017. A total of 90 seedlings were planted in each of the ten rows (N = 900). Since the spring of 2018, each month during the first growing season in the field, a seedlings height from the root collar to the apical bud was measured with an accuracy of 1 mm (see Table 1). Although the seedlings diameter is usually recognized as the single most reliable quality attribute [35,36], the seedlings height is useful in forecasting their growth and survival in the field [37,38].

Finally, before the second growing season started on 28 March 2019, we measured the height and root collar diameter of each seedling. At the same time, each seedling was photographed in the plan, from a height of 1 m using the digital camera Canon Digital IXUS 100 IS 12.1 MPix (Canon Inc., Tokyo, Japan). These photographs will be further used to create an algorithm for the automation of the monitoring process using a drone.

### 3.4. Dataset Statistical Analysis (Possible Application)

The one-way ANOVA can be used to test differences between mean values of seed germination, seedlings height and root collar diameter from different seed size and color classes. Descriptive

statistics included number of samples, mean value, standard deviation, variance, minimum value, and maximum value. Mean values could be separated using Tukey's HSD test for unequal number of samples, with a significance level of alpha = 0.05. For example, based on the data set [29], it could be assumed that the average height of seedlings from light seeds was significantly greater than from light seeds ($p = 3.32 \times 10^{-6}$). At the same time, the seed germination on the container with a dark color of the seed coat was higher than that of seeds with a light coat color ($p = 0.000013$). It is possible to use the standard chart (Box and Whisker Plot) to visualize the indicators and estimate the variability of the mean values.

## 4. User Notes

In the future, for the final validation of the Pravdin's conjecture, it is necessary to conduct research on the genetic control of seed color and its variability on the population and individual level, by use of molecular markers. Moreover, it is necessary to develop a database (Figure 1), which will set the basis for the development of the Forest Reproductive Material Library (FRMLib). To expand this library, we plan to conduct research on the "seed coat color—seedling growth" relation on other forest tree species.

Further use of this empirical dataset implies validation at the second and subsequent growing seasons. Future research will be focused on the correlation between seed coat color and a number of morphological and performance attributes of seedlings, i.e., color of needles, seedlings height, root collar diameter, seedlings crown area, etc. A repeated taking of photographs of seedlings in the plan at the end of each growing season will provide an input for automation of the reforestation success monitoring process by use of a drone.

At the same time, this study raises new questions which requires further research. For example; does the nature of distribution of seed germination parameters remain constant for seeds collected at different years and of a different origin? Does the position of a particular container in the greenhouse have an effect on seed germination? Does technology of seed grading on color reduce the genetic diversity of seedlings?

**Supplementary Materials:** The following are available online at http://www.mdpi.com/2306-5729/4/3/106/s1.

**Author Contributions:** Conceptualization, A.N. and V.I.; methodology, A.N. and V.I.; validation, A.N., V.I. and T.N.; field measurements, E.P.; formal analysis, T.N.; investigation, A.N. and E.P.; resources, A.N., E.P. and T.N.; relationship data model T.N.; data curation, A.N. and T.N.; writing—original draft preparation, A.N. and V.I.; writing—review and editing, A.N., V.I. and T.N.

**Funding:** This research received no external funding.

**Acknowledgments:** The authors acknowledge the rector of the University of Belgrade Ivanka Popović and rector of Voronezh State University of Forestry and Technologies (VSUFT) Michael Drapaluyk for the possibility of establishing scientific collaborations. The authors special gratitude is offered to the Vice-Dean for Science and International Cooperation Faculty of Forestry University of Belgrade Mirjana Šijačić-Nikolić and Vice-rector for science and innovation of VSUFT Svetlana Morkovina for scientific support and valuable comments in methodological aspects.

**Conflicts of Interest:** The authors declare no conflicts of interest.

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
