# Peer review of "Scots Pine Seedlings Growth Dynamics Data Reveals Properties for the Future Proof of Seed Coat Color Grading Conjecture"

_data_

Reviewer 1 Report

This manuscript by Novikov et al. includes unique empirical 13 data of Scots pine (Pinus sylvestris L.) container-grown seedlings outplanted in postfire site in 14 Voronezh region, Russia. These data increase our understanding of seed germination and seedling growth dynamics and color’s grading of seeds.

Below are some specific comments:

1. I did not see the statistical analysis in the data, please clarify.

2. The description of temperature condition is not correct (Line 101 and 104).

3. The sentence should be revised (Line 13-15).

4. The references should be revised to the uniform formats.

Author Response

The authors sincerely thank the reviewer for his highly professional comments, which significantly contributed to the improvement of the manuscript.

Point 1: Extensive editing of English language and style required.

Response 1: We thank the reviewer on this comment and we improved the English language and style.

Point 2: I did not see the statistical analysis in the data, please clarify.

Response 2: One of the goals of the journal, listed on the official page, is: how can data sets be used in scientific and scholarly research? We tried to stick to this goal. Therefore, we give here the appropriate, in our opinion, the possible use of this data set in statistical processing.

Separately, we point to the use of the Tukey’s HSD test as the most conservative in the study of the growth of seedlings. Bringing in the article statistical analysis of any specific parameters would lead us away from the purpose of the journal. This would be incorrect with respect to the articles we plan to write based on this dataset. However, based on your wishes, we can make some statements (see line 169-171).

Point 3: The description of temperature condition is not correct (Line 101 and 104).

Response 3: We added a description of temperature and rainfall conditions.

Point 4: The sentence should be revised (Line 13-15).

Response 4: Thanks. This sentence is revised.

Point 5: The references should be revised to the uniform formats.

Response 5: Thanks. The references revised to the uniform formats specifying a DOI (if available).

Reviewer 2 Report

In this study, correlations between seed coat color vs. seed germination rate and seedling growth in the Scots pine were analyzed, and a database was developed. The findings have important practical implications because they can contribute to increasing uniformity in seedling size, and predicting seedling growth and further development in restored forest areas. In order to optimize production in forest nurseries, i.e. promote uniform seed germination and seedling emergence, seeds can be sorted into fractions which are sown separately. The Authors decided to divide Scots pine seeds into three fractions differing in seed coat color, and two fractions differing in seed size. The question that arises here is why the fraction of seeds with a width of up to 2.5 mm (mesh sieves with circular openings were used) was considered as “non-graded” and discarded. Research shows that even within the same stand, individual trees can produce seeds of various sizes due to genetic factors. The smallest seed fraction can be discarded to eliminate empty and malformed seeds, but this can increase the risk of removing high-quality seeds which are naturally smaller in size that the average value for the species. It appears that the Authors have also noted this limitation of the study, as suggested by the last question in the manuscript, although this question could also refer to the transfer of the gene pool by only one fraction of Scots pine seeds.

The data are original, and their source is well defined, although the scope of the study is rather narrow and local because the seeds were collected from one region only. The metadata well describe the research data, although some more details are needed. The size groups of seeds should be clearly described, stating that the group of “small” seeds comprises seeds whose width ranges from 2.51 to 3.25 mm, and that the group of “large” seeds comprises seeds whose width exceeds 3.25 mm. This information should also be given in Table 2. In the header row of Table 3, phrases such as “Light 30” should be replaced with “Light seeds, day 30”, etc. for greater clarity. In the Excel file, in the “container soil germination” card, not all columns contain the relevant statistical parameters (what does “MGT” stand for? This parameter was not explained in the text, and it is not given for all groups – why?).

The manuscript is generally well-organized and formatted. Please use the appropriate symbol in “°C” instead of the digit zero (0) in the superscript (lines 101 and 104). Do not insert a space between the percentage value and the symbol “%” (line 101). The entries in the Reference section have double numbers – please correct, and include DOI numbers (if available).

Author Response

We thank the reviewer for the valuable comment and provided the suggested additions in the revised version.

Point 1: In this study, correlations between seed coat color vs. seed germination rate and seedling growth in the Scots pine were analyzed, and a database was developed. The findings have important practical implications because they can contribute to increasing uniformity in seedling size, and predicting seedling growth and further development in restored forest areas. In order to optimize production in forest nurseries, i.e. promote uniform seed germination and seedling emergence, seeds can be sorted into fractions which are sown separately. The Authors decided to divide Scots pine seeds into three fractions differing in seed coat color, and two fractions differing in seed size. The question that arises here is why the fraction of seeds with a width of up to 2.5 mm (mesh sieves with circular openings were used) was considered as “non-graded” and discarded. Research shows that even within the same stand, individual trees can produce seeds of various sizes due to genetic factors. The smallest seed fraction can be discarded to eliminate empty and malformed seeds, but this can increase the risk of removing high-quality seeds which are naturally smaller in size that the average value for the species. It appears that the Authors have also noted this limitation of the study, as suggested by the last question in the manuscript, although this question could also refer to the transfer of the gene pool by only one fraction of Scots pine seeds.

Response 1: You are right: the number of fractions in the division of seeds is a rather controversial topic. We were based on our own research conducted with seedlings of seeds, sorted only by size without color. This study showed when sorting by size to divide the seeds into only two fractions. We have clarified this (see line 75). However, we believe that it is necessary to use valuable seed material as fully as possible, so we will continue to explore new ways of improvement for sorting.

Point 2: The data are original, and their source is well defined, although the scope of the study is rather narrow and local because the seeds were collected from one region only. The metadata well describe the research data, although some more details are needed. The size groups of seeds should be clearly described, stating that the group of “small” seeds comprises seeds whose width ranges from 2.51 to 3.25 mm, and that the group of “large” seeds comprises seeds whose width exceeds 3.25 mm. This information should also be given in Table 2.

Response 2: Thanks. We described this issue better.

Point 3: In the header row of Table 3, phrases such as “Light 30” should be replaced with “Light seeds, day 30”, etc. for greater clarity.

Response 3: Done.

Point 4: In the Excel file, in the “container soil germination” card, not all columns contain the relevant statistical parameters (what does “MGT” stand for? This parameter was not explained in the text, and it is not given for all groups – why?)

Response 4: Mean germination time (MGT), the average time required for seeds to germinate, was determined using the formula (Ellis and Roberts 1980a).

This parameter is not widely studied for container nurseries. Therefore, we have given it for information in the revised Excel file and do not see the feasibility of its calculation for all groups and inclusion in the manuscript. We highlighted it in red with a mark (for information).

Point 5: The manuscript is generally well-organized and formatted. Please use the appropriate symbol in “°C” instead of the digit zero (0) in the superscript (lines 101 and 104). Do not insert a space between the percentage value and the symbol “%” (line 101).

Response 5: Done.

Point 6: The entries in the Reference section have double numbers – please correct, and include DOI numbers (if available)

Response 6: Thanks. We have added to the Reference section of DOI numbers.  

This manuscript is a resubmission of an earlier submission. The following is a list of the peer review reports and author responses from that submission.